| Antimicrobial Chemotherapy | Research Article

# Effective killing of *Mycobacterium abscessus* biofilm by nanoemulsion delivery of plant phytochemicals

Casey Albano,[1] Ahmed Nabawy,[2] Wyatt C. Tran,[1] Malavika Prithviraj,[1] Takehiro Kado,[1] Muhammad Aamir Hassan,[2] Jessa Marie V. Makabenta,[2] Vincent M. Rotello,[2] Yasu S. Morita[1]

**ABSTRACT** *Mycobacterium* is an acid-fast, aerobic, non-motile, and biofilm-forming bacterium. The increasing prevalence of mycobacterial infections makes it necessary to find new methods to combat the resistance of bacteria to conventional antibiotics. *Mycobacterium abscessus* is an emerging pathogen that is intrinsically drug resistant due to several factors, including an impermeable cell envelope, drug efflux pumps, target-modifying enzymes, and the ability to form thick, robust biofilms. Phytochemicals are promising antimicrobials; however, their poor solubility in water and their inability to penetrate biofilms render them inefficient in killing bacterial biofilms. In this study, we demonstrate the efficacy of polymer-stabilized phytochemical nanoemulsions in killing *M. abscessus* biofilms. These nanoemulsions improve the solubility and stability of the phytochemicals and enable biofilm penetration and eradication. We show that the phytochemical emulsions effectively eliminated *M. abscessus* in an *in vitro* biofilm model and killed non-replicating persister cells in the Wayne hypoxia model. These nanoemulsions were also effective *in vivo* in a wound infection model. These findings demonstrate the potential of polymer-stabilized phytochemical nanoemulsions as a promising alternative to conventional antibiotics for the treatment of mycobacterial infections.

**IMPORTANCE** *Mycobacterium abscessus* is among the opportunistic bacterial pathogens that cause nontuberculous mycobacterial diseases. The infection caused by *M. abscessus* is difficult to treat because the bacterium is resistant to many of the currently available antibiotics, limiting chemotherapeutic strategies. Furthermore, it forms biofilms in clinically relevant settings, making the infection difficult to treat. Many phytochemicals have potent antimicrobial activities, but their hydrophobicity limits clinical applications. In this study, we tested a new drug delivery strategy where hydrophobic plant phytochemicals were emulsified with a biodegradable nanosponge. We show that the emulsification makes phytochemicals such as carvacrol and eugenol more effective against *M. abscessus* biofilms. We further demonstrate that nanoemulsified phytochemicals can kill hypoxia-induced dormant *M. abscessus* and effectively improve skin wound infection in mice. Our data pave the way to use phytochemical nanosponge as a platform to create synergy by combining other antimycobacterial drugs.

**KEYWORDS** antimicrobial agents, essential oils, drug delivery, *Mycobacterium*, nanoemulsion, phytochemical

*M*ycobacterium abscessus is a nontuberculous mycobacterium that is emerging as a highly drug-resistant, rapidly growing, opportunistic pathogen (1). *M. abscessus* infection is difficult to treat due to its impermeable cell envelope, drug efflux pumps, and target-modifying enzymes that make the pathogen intrinsically resistant to antibiotics (2). Mycobacteria have a distinct outer membrane (OM) that is rich in

**Peer Reviewer** John Jairo Aguilera-Correa, IIS-Fundacion Jimenez Diaz, Madrid, Spain

Address correspondence to Yasu S. Morita, ymorita@umass.edu.

Casey Albano and Ahmed Nabawy contributed equally to this article. Author order was determined by the leadership role taken for this project.

The authors declare no conflict of interest.

See the funding table on p. 12.

unique lipids, such as trehalose dimycolates, glycopeptidolipids, phthiocerol dimycocerosates, and phenolic glycolipids (3–6). These OM lipids form a potent barrier, making the cell envelope impermeable to small molecule antibiotics. Indeed, the cell envelope of mycobacteria is 100–1,000-fold less permeable than that of other bacteria like *Escherichia coli* and *Pseudomonas aeruginosa* against hydrophilic solutes (7). The inefficient permeation across the OM is hypothesized to account for much of the intrinsic resistance of mycobacteria against antibiotics.

In addition to molecular defenses, *M. abscessus* can produce extracellular polymeric substances (EPS) to create thick, robust biofilms that contribute to antibiotic resistance and virulence (8–13). *M. abscessus* biofilms contaminate drinking water and medical devices (14–16) and are found in human patients with pulmonary *M. abscessus* infections (17, 18). The polysaccharides, proteins, and extracellular DNA create an EPS matrix that protects resident cells from antibiotics (5, 8, 9, 13, 19). Mycobacterial biofilms also contain mycolic acids as a lipid component (20, 21). Recent studies suggest that *M. abscessus* biofilm produces EPS composed of lipids, proteins, carbohydrates, and extracellular DNA (12), and when grown in medium mimicking cystic fibrosis sputum, extracellular DNA, mannose- and glucose-containing glycans, and phospholipids were the major EPS components (11).

The treatment for *M. abscessus* infection is usually a combination of oral macrolide-based therapy with additional intravenous drugs for at least 2 weeks. Common treatment is a cocktail of clarithromycin, amikacin, and cefoxitin (22, 23). Other antimicrobials such as tigecycline, linezolid, and imipenem have been used as well. These antibiotics often induce adverse side effects (24) and have led to the emergence of drug-resistant mutants in *in vitro* experiments and patients (25–30). New treatments are urgently needed as treatments for *M. abscessus* pulmonary infections have poor success rates of less than 50% (31–34).

While chronic pulmonary infection is the most common current clinical manifestation (22), *M. abscessus* can infect any tissue (35) and is one of the most common rapidly growing mycobacteria found in the wound and cutaneous infections (36). Direct contact or puncture wounds, such as those from surgical operations, tattooing, and body piercing, can lead to skin or soft tissue infections (37), and one study suggests a minimum of 3-month antibiotic treatment for iatrogenic *M. abscessus* infection transmitted by acupuncture (38).

Plant essential oils contain hydrophobic phytochemicals produced by plants as a part of their host immune system (39, 40). Phytochemicals such as carvacrol and eugenol have broad-spectrum antimicrobial activity and show bactericidal effects against *M. abscessus* and other mycobacteria (41–45). The bactericidal effects are thought to be mediated, at least in part, through damaging bacterial cytoplasmic membranes (46). The hydrophobic nature of phytochemicals poses a challenge for essential oil-based treatment of biofilm-based infections due to their lack of solubility in aqueous environments and inefficient penetration into biofilms (47, 48).

Polymer-stabilized essential oil nanoemulsions present a promising strategy for combating bacterial infections (46, 49–52). We have developed a strategy for creating these nanoemulsions by dissolving a biodegradable crosslinker in an antimicrobial essential oil (Fig. 1A) and then emulsifying with a polymer in water (Fig. 1B). The result is a biodegradable polymeric nanoemulsion (BNE) that increases the solubility of hydrophobic essential oils and antibiotics (53–56). These BNEs have unique physicochemical properties that enable biofilm penetration (51). BNE can effectively be delivered to bacteria within an EPS matrix, and once BNE encounters bacterial membranes, the encapsulated oil diffuses into bacterial membrane in a contact-dependent fashion (Fig. 1C). These systems have demonstrated high efficacy against biofilms *in vitro* and *in vivo* in wound biofilm models.

Previous studies using BNEs have focused on *Staphylococcus aureus* and related pathogens. In this study, we aimed to test the effect of BNE-mediated antimicrobial delivery against mycobacteria. We demonstrate the efficacy of the BNEs against atypical

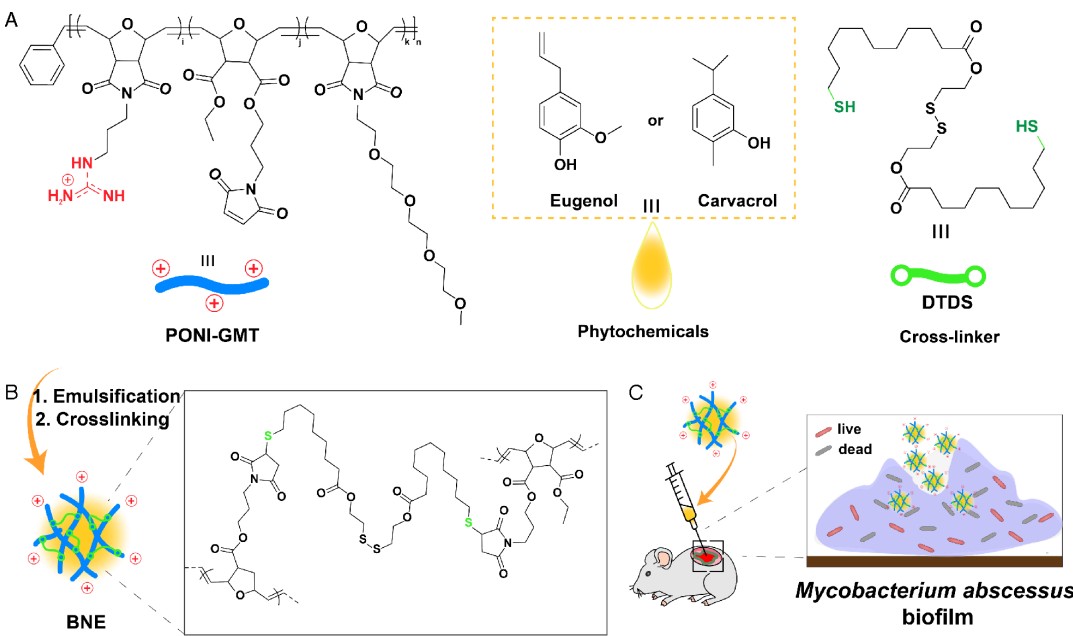

**FIG 1** Fabrication scheme of biodegradable polymeric nanoemulsions incorporating carvacrol and eugenol. (A) Structures of poly(oxanorborneneimide) scaffold bearing guanidine, maleimide, and tetraethyleneglycol monomethyl ether groups (PONI-GMT), phytochemicals (eugenol and carvacrol), and biodegradable dithiol-disulfide (DTDS). (B) Process for emulsification and crosslinking that enhances the stability of the nanoemulsions. (C) Schematic representation of the efficient penetration of BNEs into wound biofilm murine model, leading to the effective killing of residing bacteria.

and refractory *M. abscessus* biofilms *in vitro* and in wound biofilm infections in a mouse model. Our results show that BNE-emulsified essential oils can effectively kill *M. abscessus* even under persistent biofilm growth or oxygen-depleted dormancy model and offer a new platform to encapsulate anti-mycobacterial drugs for targeted antibiotic delivery.

## RESULTS AND DISCUSSION

The BNEs used in the current studies employ poly(oxanorborneneimide) scaffold-bearing guanidine, maleimide, and tetraethyleneglycol monomethyl ether groups (PONI-GMT) (53–56). This positively charged polymer platform interacts with negatively charged bacterial surface structures and facilitates penetration into biofilms. We previously demonstrated that nanoemulsion is effective in killing various biofilm-forming pathogens, such as *Klebsiella pneumoniae*, *P. aeruginosa*, *Acinetobacter baumannii,* and methicillin-resistant *Staphylococcus aureus* (MRSA) (54). Tetraethyleneglycol monomethyl ether sidechains confer amphiphilicity to the polymers, facilitating the emulsification of the essential oil. The PONI-GMT polymers were stabilized by a biodegradable dithiol-disulfide (DTDS) crosslinker, encapsulating a hydrophobic essential oil (Fig. 1A and B). We fabricated BNEs using our established protocol (Fig. 1A and B) (53–56). Dynamic light scattering (DLS) measurements revealed an average size of 250 nm for C-BNE and 200 nm for E-BNE, respectively (Fig. 2A). Due to the guanidinium group in PONI-GMT, the zeta potential of BNEs was around +20 mV, indicating a positively charged surface (Fig. 2B). Morphological analysis of BNEs using transmission electron microscopy supported the size of BNEs determined by DLS (Fig. 2C).

We first tested the BNEs against *Mycobacterium smegmatis* as it is an established model *Mycobacterium* capable of forming robust biofilms and a useful platform for drug discovery (57). We grew *M. smegmatis* cultures planktonically to the log phase and diluted them to an $OD_{600}$ of 0.1. The medium was then supplemented with 8% (vol/vol) carvacrol or eugenol-loaded biodegradable polymeric nanoemulsions (C-BNE or E-BNE). As a control, we used 3.1 mM carvacrol or 3.1 mM eugenol, which is equivalent to the concentration of the phytochemical present in the final 8% C-BNE or

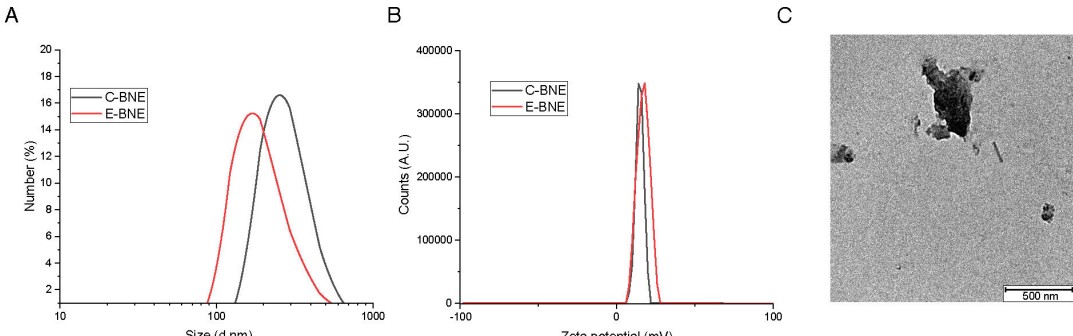

**FIG 2** Characterization of biodegradable nanoemulsion. (A) DLS histogram of C-BNE and E-BNE. (B) Charge characterization of C-BNE and E-BNE through zeta potential, respectively. (C) Transmission electron microscopy images of 12% E-BNE. Due to the evaporation of the oil component during the sample preparation, the nanoemulsion appears irregular in shape under electron microscopy.

E-BNE. Colony-forming units (CFU) were determined every 12 hours for 48 hours. *M. smegmatis* sustained viability for the first 12 hours of 3.1 mM carvacrol treatment, but the CFU declined logarithmically beyond this initial persistent phase. C-BNE was more effective against planktonic *M. smegmatis* than carvacrol alone (Fig. 3A), suggesting that localized nanoemulsion-mediated delivery makes carvacrol more effective. The eugenol nanoemulsion (E-BNE) yielded more striking results. Unlike carvacrol, eugenol alone was not bactericidal and only mildly bacteriostatic against *M. smegmatis* (Fig. 3B). In contrast, E-BNE eradicated *M. smegmatis* within 48 hours (Fig. 3B). These results demonstrate that the bactericidal effect of eugenol emulsified with nanoemulsions is markedly enhanced in planktonically grown *M. smegmatis*.

BNEs are particularly effective against bacterial biofilms (51, 53, 54), and we next tested their effect on *M. smegmatis* biofilm. We grew pellicles (a liquid surface biofilm) for 5 days and applied either carvacrol or eugenol and their corresponding BNEs. *M. smegmatis* formed robust pellicles. Replacement of the medium with a BNE-containing medium transiently disrupted the pellicle, but the overall appearance of the pellicle did not change significantly with or without BNE treatments (Fig. S1). C-BNE was strikingly effective against *M. smegmatis* biofilm, completely eradicating cells in 24 hours (Fig. 3C). While E-BNE was less effective, it was more effective than the treatment with eugenol alone (Fig. 3D). These data together indicate that nanoemulsions are significantly more effective against *M. smegmatis* than their constituent essential oils.

Given the promising bactericidal effects of BNEs on *M. smegmatis*, we next explored their impact on *M. abscessus* in detail, starting with the effects of plant phytochemicals on *M. abscessus* plasma membrane. Many phytochemicals have membrane-fluidizing properties (58). In mycobacteria, membrane fluidization induces fatty acid remodeling of phosphatidylinositol mannosides (PIMs), in which a fourth fatty acid is added to the inositol moiety (Fig. 4A) (59). At 1.6 and 3.1 mM, both carvacrol and eugenol induced PIM inositol acylation in *M. abscessus* (Fig. 4B). Carvacrol was more potent than eugenol, and for both phytochemicals, inositol acylation of AcPIM6 was more prominent than that of AcPIM2. We next tested the effect of these phytochemicals on cell envelope permeability. We incubated planktonic *M. abscessus* cells at 37°C for 15 min with TO-PRO-3, a membrane-impermeable DNA staining dye, in the presence of 256 mM dimethyl sulfoxide (DMSO), 100 mM benzyl alcohol, 3.1 mM carvacrol, or 3.1 mM eugenol and analyzed TO-PRO-3 fluorescence by flow cytometry. While DMSO had no effect, membrane fluidizer benzyl alcohol made ~90% of cells stained by TO-PRO-3 (Fig. 4C). Carvacrol and eugenol were also effective, making ~90% of cells TO-PRO-3-positive (Fig. 4C). Taken together, these results suggest that phytochemicals fluidize membrane and make cell envelope more permeable to small hydrophilic molecules.

Despite the permeabilizing effect of phytochemicals on cell envelope, planktonically grown *M. abscessus* sustained their viability for at least 12 hours in the presence of carvacrol. However, viability declined after the initial persistent phase (Fig. 5A). Similar to

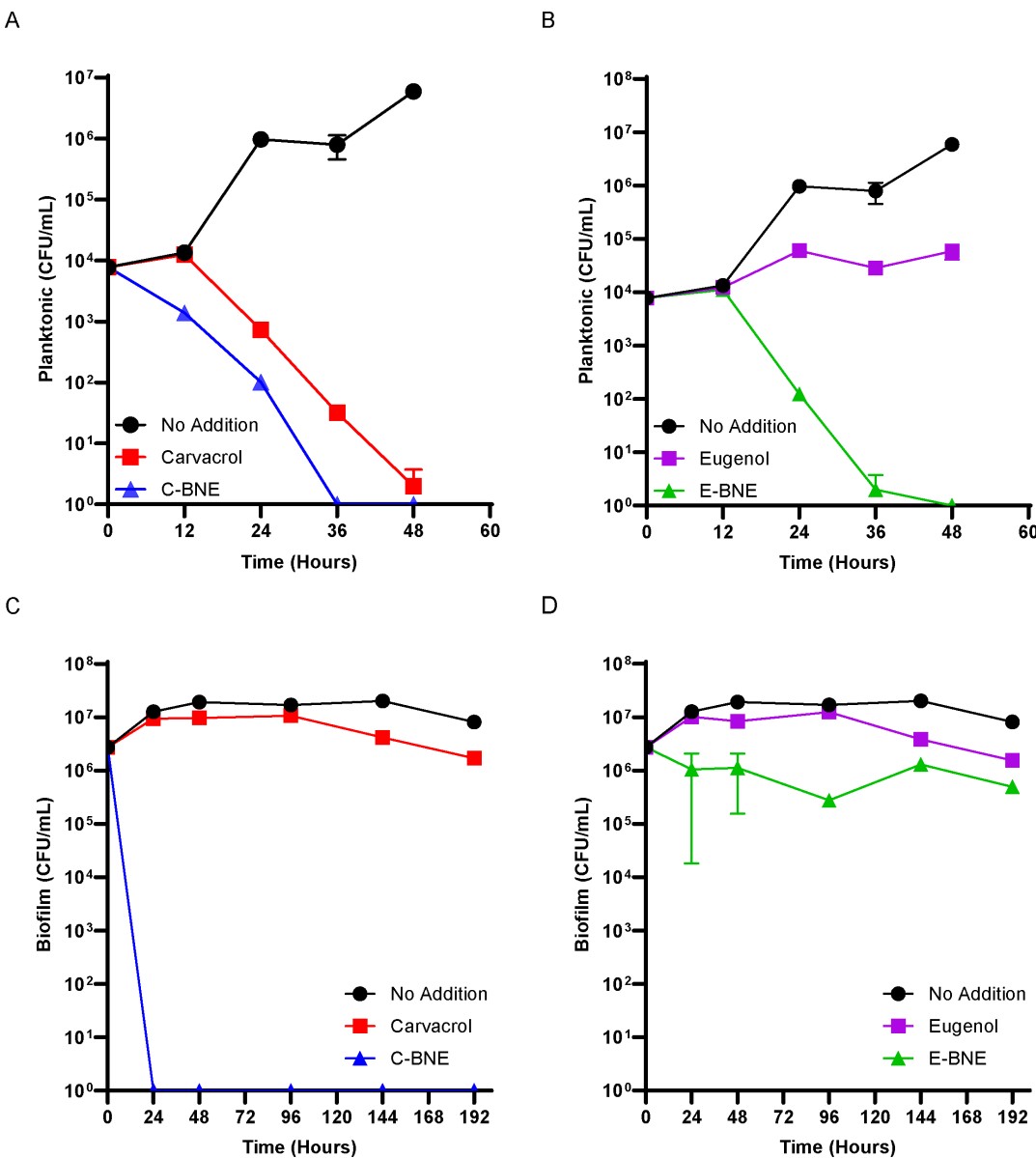

**FIG 3** Antimycobacterial effects of nanoemulsion-encapsulated phytochemicals on planktonic and biofilm *M. smegmatis. M. smegmatis* planktonic cultures were grown to the log phase ($OD_{600}$ = 0.6–0.8) and diluted back to an $OD_{600}$ of 0.1. Pellicle biofilm was grown for 5 days before initiating the antimycobacterial treatments. (A) Planktonic *M. smegmatis* treated with 3.1 mM carvacrol or 8% (vol/vol) C-BNE (3.1 mM carvacrol equivalent). (B) Planktonic *M. smegmatis* treated with 3.1 mM eugenol or 8% (vol/vol) E-BNE (3.1 mM eugenol equivalent). (C) *M. smegmatis* biofilm treated with 3.1 mM carvacrol or 8% (vol/vol) C-BNE. (D) *M. smegmatis* biofilm treated with 3.1 mM eugenol or 8% (vol/vol) E-BNE. Time 0 hour indicates when phytochemicals were added. Experiments were done in triplicate, and averages and standard deviations are shown. Note that phytochemicals are essential components of BNE preparations, and thus we cannot test a control BNE that does not contain phytochemicals.

the observations in *M. smegmatis*, nanoemulsion formulation, C-BNE, had little additional effect on the bactericidal effect of carvacrol. In contrast, eugenol oil alone was ineffective against *M. abscessus,* and the cells grew although it was slower than the no treatment control (Fig. 5B). Notably, E-BNE was bactericidal to *M. abscessus* cells in contrast to the ineffective eugenol.

Carvacrol and C-BNE were equally effective against planktonic *M. smegmatis* and *M. abscessus*, although there was a slight delay in the effect of carvacrol against *M. smegmatis* in comparison to C-BNE (see Fig. 3A). These results were expected as phytochemicals can reach its membrane targets easily in the case of planktonic

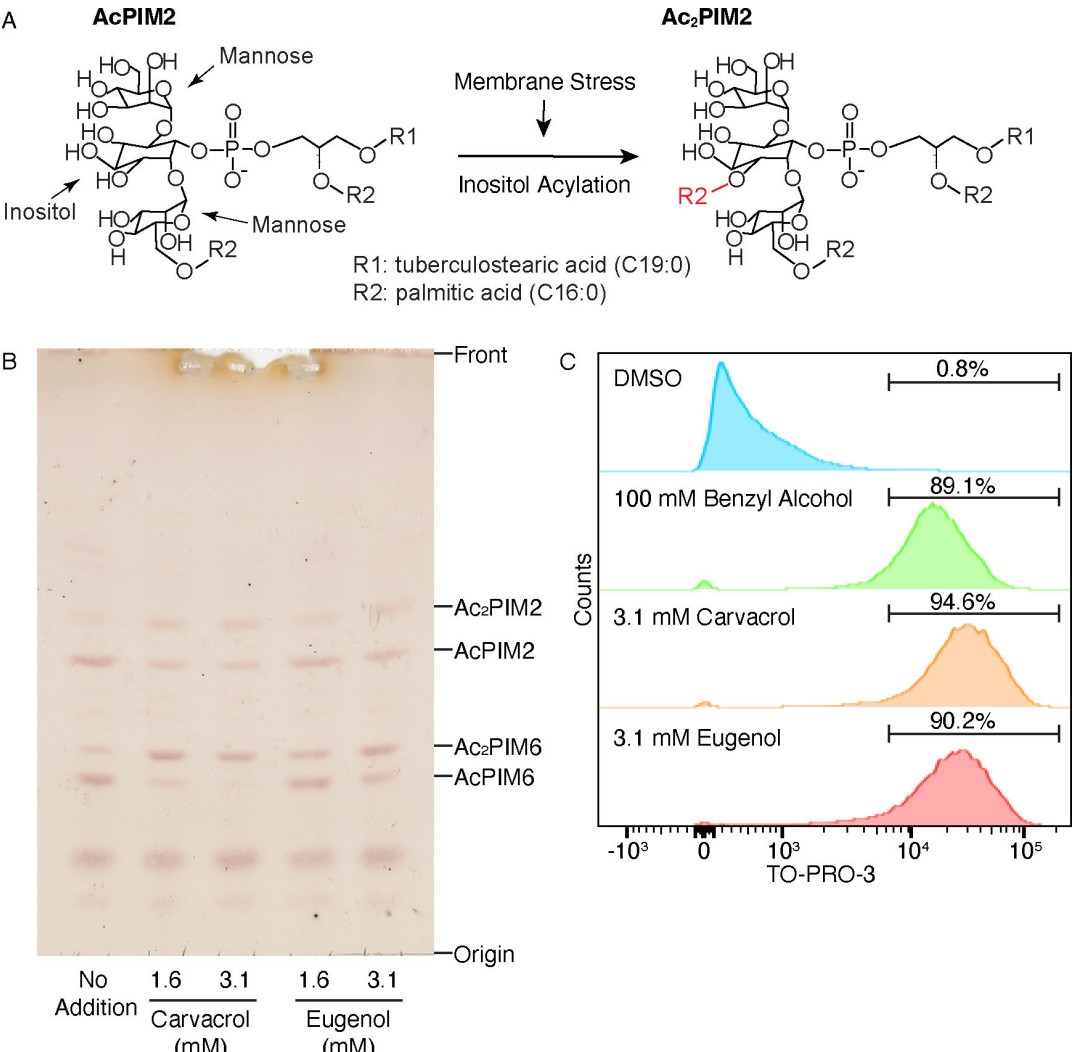

**FIG 4** PIM inositol acylation and increased membrane permeability in *M. abscessus* induced by phytochemicals. (A) The reaction of PIM inositol acylation. A fatty acid is added to the 3-OH of *myo*-inositol in response to membrane fluidization stress. R1 and R2 are tuberculostearic acid and palmitic acid, respectively, in *M. smegmatis*, but they have not been determined in *M. abscessus*. (B) High-performance thin layer chromatography analysis of PIM inositol acylation in response to 1.6 and 3.1 mM of carvacrol or eugenol. (C) Flow cytometry results of *M. abscessus* treated with DMSO (negative control), 100 mM benzyl alcohol, 3.1 mM carvacrol, and 3.1 mM eugenol. Membrane permeability was determined using TO-PRO-3, a membrane-impermeable fluorescent DNA staining dye.

cells. Therefore, it was surprising that E-BNE was substantially more effective than the equivalent concentration of eugenol alone against both planktonic *M. smegmatis* and *M. abscessus*. In fact, eugenol showed only mild bacteriostatic activities against both *M. abscessus* and *M. smegmatis* planktonic cells at the concentration tested (3.1 mM = 508 µg/mL). This is in line with previous findings that the MIC value of eugenol is >100 µg/mL against *Mycobacterium tuberculosis* (44), while the MICs of carvacrol against rapidly growing mycobacteria are ~64 µg/mL (43). Since mycobacteria form small granules even when they are growing planktonically (60), we speculate that nanoemulsions may have a minor but significant effect against planktonically grown cells.

*M. abscessus* is commonly found in hypoxic microenvironments, including viscous mucus of the lungs of cystic fibrosis patients (61, 62), soft tissues (63), and macrophages (64). Under hypoxic conditions, *M. abscessus* shows enhanced tolerance to antibiotics (65). We, therefore, tested the effect of the nanoemulsions against cells under hypoxia in the Wayne hypoxia model (66). *M. abscessus* were grown to a log phase, placed under a hypoxic condition for 5 days, and exposed to phytochemical and antibiotic treatments.

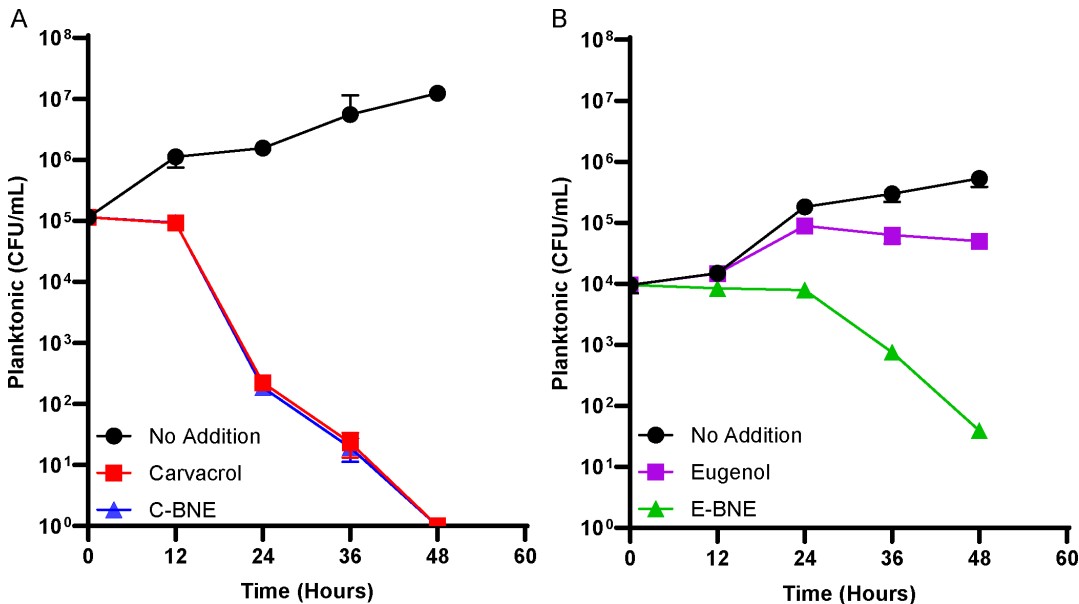

**FIG 5** Nano-emulsified phytochemicals have bactericidal effects on planktonically grown *M. abscessus*. (A) *M. abscessus* cultures were treated with no addition, 8% (vol/vol) C-BNE, or 3.1 mM carvacrol. (B) *M. abscessus* cultures were treated with no addition, 8% (vol/vol) E-BNE, or 3.1 mM eugenol. Time 0 indicates the onset of the treatments. All treatments were performed in triplicate, and standard deviations are shown.

Linezolid is one of the drugs clinically used to treat *M. abscessus* infection (67) and has been suggested to be bactericidal against nonreplicating persister *Mycobacterium tuberculosis* (68). Under hypoxic conditions, linezolid showed only a small decline in CFU after 48 hours of drug exposure (Fig. 6). While eugenol alone had little effect, E-BNE decreased the viability of hypoxic *M. abscessus* by two logs (Fig. 6), indicating the effectiveness of nanoemulsified eugenol against refractory hypoxic *M. abscessus*.

Since *M. abscessus* infection is commonly a biofilm disease (15, 17, 18), we next examined how effectively BNEs penetrated *M. abscessus* biofilms and eradicated cells within the biofilm structure. We first incubated 7-day-old pellicle of *M. abscessus* with Nile Red-loaded E-BNE. *M. abscessus* was labeled with SYTO9 green fluorescent DNA staining dye. We visualized the distribution of E-BNE using confocal microscopy. E-BNE penetrated through the biofilm within 3 hours, as shown with the colocalization of both dyes (Fig. 7A). These results indicate that cationic and amphiphilic properties of nanoemulsion scaffold allow effective penetration of encapsulated hydrophobic molecules through *M. abscessus* biofilms, as we have previously observed for other bacterial biofilms.

Next, we tested the bactericidal effects of carvacrol and eugenol BNEs against the biofilms of *M. abscessus*. We applied phytochemicals with or without nanoemulsion to a 7-day-old biofilm and examined the CFU for up to 7 days. *M. abscessus* pellicles were not as robust as those of *M. smegmatis* but similar to *M. smegmatis*; the overall appearance of pellicles did not change significantly with or without BNE treatments (Fig. S1B). In contrast to planktonic cells (see Fig. 5), carvacrol was not effective in killing *M. abscessus* biofilm (Fig. 7B). Strikingly, emulsified carvacrol (C-BNE) killed *M. abscessus* in 6 days (Fig. 7B). Consistent with the observations with planktonic cells, eugenol was less effective in killing *M. abscessus* than carvacrol but emulsified eugenol (E-BNE) reduced CFU by two log units in 6 days (Fig. 7C). It took longer to kill *M. abscessus* biofilms than planktonic cells. It is likely that BNE penetration of the biofilm is slower than its interactions with planktonic bacteria. Notably, the polymer is in low concentrations in the nanoemulsion (330 µg/mL of 100% BNE), and the BNE is likewise in low concentration relative to other nutrients in the media. Therefore, the slow killing is unlikely to be due to the polymer serving as a growth-promoting nutrient source. Importantly, while mycobacterial biofilm EPS is known to be lipid rich, the fact that emulsifying lipophilic

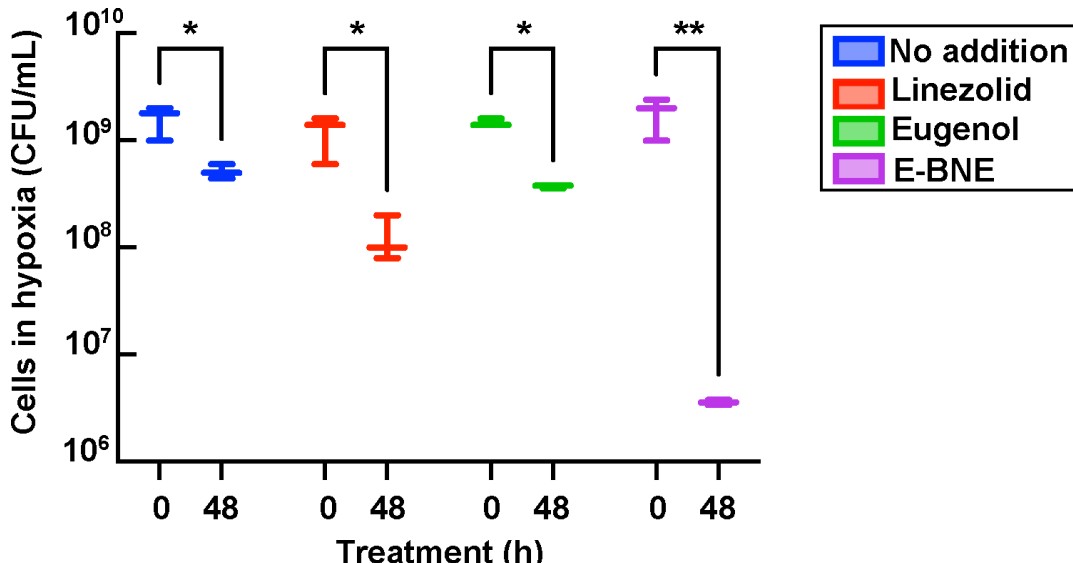

**FIG 6** Bactericidal effect of nanoemulsion-encapsulated phytochemicals on *M. abscessus* cells under hypoxia. Hypoxic *M. abscessus* cells treated with 28.8 µg/mL linezolid or 3.1 mM eugenol or 8% E-BNE. Time 0 hour indicates when phytochemicals or drugs were added. Experiments were done in triplicate. The bold central line represents the mean, with error bars indicating the standard deviation. Statistical significance was determined by two-way ANOVA, followed by Šídák's multiple comparisons test. *$P < 0.05$ and **$P < 0.005$.

phytochemicals improved their efficacy suggests that the EPS is sufficiently hydrophilic to allow the penetration of hydrophilic nanoemulsified particles.

We finally tested the *in vivo* efficacy of E-BNE as a topical wound biofilm therapeutic. In both *M. smegmatis* and *M. abscessus*, the bactericidal effect of E-BNE was not as robust as C-BNE. However, we have previously shown that E-BNE is less toxic to 3T3 fibroblast cells than C-BNE (55). Therefore, we used E-BNE to test its efficacy against *M. abscessus* in a wound infection model. Our studies are based on a robust murine model of severe wound biofilm infection previously established for Gram-positive bacterial infections such as MRSA (54). We first created a wound on the dorsum of the mice using a 5-mm skin punch, infected the wound with $10^8$ CFU of luciferase-expressing *M. abscessus* (69), and incubated for 4 days to develop a biofilm. After 4 days, three groups of three mice were randomized to receive one of the following topical treatments: (i) PBS, (ii) Linezolid, or (iii) E-BNE. Treatments were done once daily for 3 days (Fig. 8A). The *M. abscessus* infection persisted for 7 days in PBS-treated mice (control), and topical application of linezolid was only mildly effective as visualized by luciferase luminescence (Fig. 8B) and CFU (Fig. 8C). In contrast, mice treated with E-BNE showed substantial reductions in bacterial burden, demonstrated by barely visible luminescence (Fig. 8B) and ~1.5 $\log_{10}$ reduction in CFU (Fig. 8C). CFU is generally more sensitive than In Vivo Imaging System (IVIS) imaging (70), but for confirmatory purposes, we analyzed the images taken from IVIS imaging system through ImageJ platform (71). The region of interest was selected and examined by integrated density analysis (Fig. S2). The overall patterns were similar to those observed by CFU. These data demonstrate the effectiveness of BNE-based delivery in a wound infection setting, which is a relevant model as there are clinical cases of skin and soft tissue infections of nontuberculous mycobacteria, including *M. abscessus* (72).

In summary, our current study demonstrated that the cationic and amphiphilic PONI-based polymeric scaffold that is effective in penetrating other bacterial biofilms was highly effective against the biofilms of both *M. smegmatis* and *M. abscessus*, underscoring the feasibility of nanoparticle-based drug delivery approach against recalcitrant mycobacterial biofilm infections. We tested a smooth morphotype of *M. abscessus* in this study. *M. abscessus* rough morphotypes also form biofilms (73, 74), and

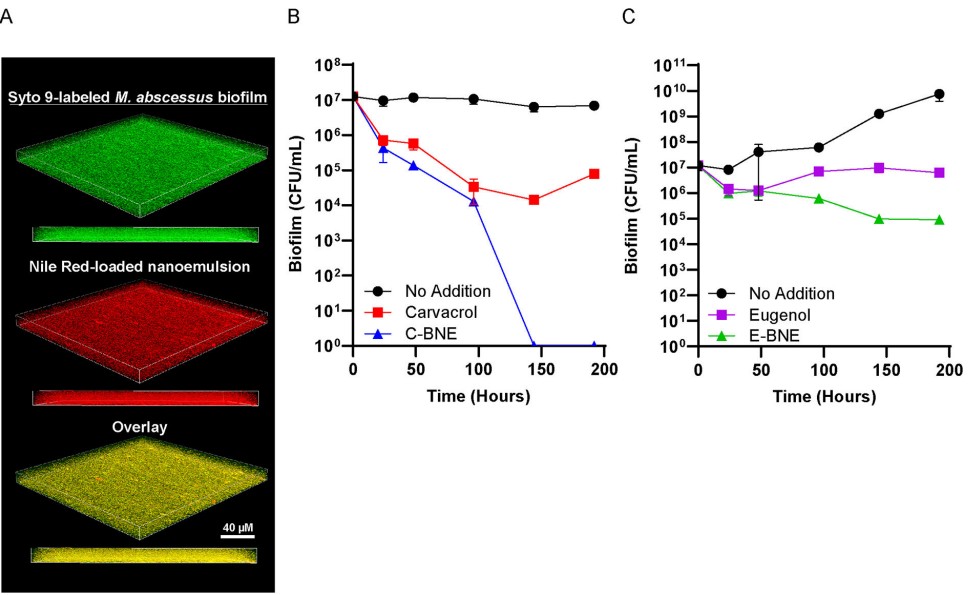

**FIG 7** Effect of nanoemulsions on *M. abscessus* biofilm. (A) *M. abscessus* biofilm was stained with SYTO 9 green fluorescent DNA stain and incubated with Nile Red-loaded E-BNE for 3 hours. Fluorescence was visualized by confocal fluorescence microscopy. (B) *M. abscessus* biofilms were grown for 7 days in M63 medium. The biofilm was treated with no addition, 8% (vol/vol) C-BNE, and 3.1 mM carvacrol dissolved in M63. (C) *M. abscessus* biofilms were grown for 7 days in M63 medium. The biofilm was treated with no addition, 8% (vol/vol) E-BNE, and 3.1 mM eugenol dissolved in M63. All treatments were performed in triplicate, and standard deviations are shown.

the effects of BNE-mediated compound delivery on rough morphotypes remain to be tested.

The generation of these BNEs is scalable, making them a promising platform for translation to the clinic. In future studies, we envision utilizing E-BNE as a medium to co-emulsify additional anti-mycobacterials. Several currently used antimycobacterial drugs are highly lipophilic, making them difficult to deliver to the bacteria protected by EPS and multilayered cell envelope. Such combinations of a plant phytochemical and anti-mycobacterials may create synergy to kill *M. abscessus*. Our wound infection model would be useful for evaluating the therapeutic efficacy and drug delivery methodologies against *M. abscessus* infection in future studies.

## MATERIALS AND METHODS

### Preparation and characterizations of nanoemulsions

BNE was prepared through emulsification of essential oil, either eugenol or carvacrol, into an aqueous PONI-GMT solution. Briefly, DTDS (3%, wt/vol) was solubilized in the essential oil, and 3 µL of the oil mixture was added to the PONI-GMT aqueous solution (497 µL, 6 µM). This solution was then emulsified for 50 seconds using an amalgamator. Emulsions were allowed to rest overnight prior to use.

### Planktonic culture and antimycobacterial sensitivity assay

All planktonic cultures of *M. smegmatis* mc$^2$155 and *M. abscessus* ATCC19977 (smooth morphotype) were grown in Middlebrook 7H9 broth supplemented with 15 mM NaCl, 0.2% (wt/vol) glucose, and 0.05% (vol/vol) Tween-80, shaking at 120 rpm at 37°C. Primary cultures were diluted to an $OD_{600}$ of 0.1 and mixed with eugenol, carvacrol, C-BNE, and E-BNE as described below to initial experimental cultures. Eugenol and carvacrol stocks (2× concentrated) were freshly prepared at 1,016 and 938 µg/mL in Middlebrook 7H9, respectively. Nanoemulsion (C-BNE or E-BNE) stocks (2× concentrated) were freshly

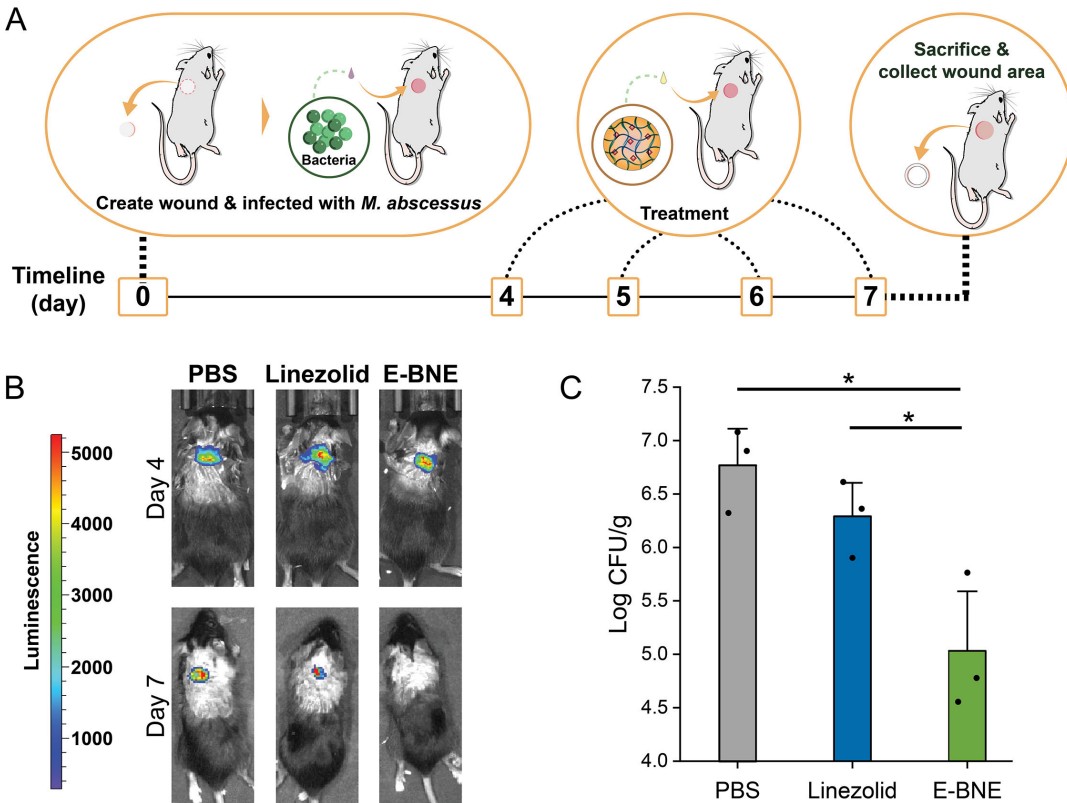

**FIG 8** *M. abscessus* skin infection model. (A) Experimental procedure. $10^8$ CFU of luciferase-expressing *M. abscessus* was inoculated. Treatment was either 85 µM linezolid or E-BNE (containing 3.1 mM eugenol). (B) Luciferase luminescence was measured by an IVIS spectrum-CT imaging system. (C) At day 7, mice were sacrificed and CFU per gram skin lesion was measured in triplicate, with error bars indicating standard deviation. *$P < 0.05$ determined by one-way ANOVA and Newman-Keuls multiple comparison test.

prepared at 16% (vol/vol) in the same medium. For each treatment, an equal volume of culture ($OD_{600} = 0.1$) and medium supplemented with a phytochemical were mixed to achieve a final concentration of 508 µg/mL eugenol or 469 µg/mL carvacrol (both at 3.1 mM) or 8% BNE, which is equivalent of 3.1 mM phytochemical. All experimental cultures were prepared in triplicate. *M. smegmatis* cultures were grown in culture tubes (16 × 125 mm), while *M. abscessus* cultures were grown in inkwell bottles (Nalgene, 30 mL).

## Biofilm culture and antimycobacterial sensitivity assay

*M. abscessus* biofilms were prepared from frozen stock of planktonically grown cells (1.13 × $10^9$ CFU/mL), which was thawed completely and directly inoculated into M63 medium to achieve a starting density of $5.0 × 10^5$ CFU/mL. A 500 µL aliquot of cell suspension was placed in an autoclaved 1.7 mL microtube that was punctured on the lid with a thumb tack (1 mm diameter) to allow gas exchange. The tube was incubated at 37°C for 7 days to form a pellicle. At day 7, 250 µL of the medium was carefully removed by a Pipetman and replaced with M63 medium containing carvacrol or eugenol to achieve the final concentrations of 508 or 469 µg/mL, respectively. For nanoemulsion treatment, C-BNE or E-BNE was prepared at 16% (vol/vol) in M63 medium, and 250 µL of the medium was replaced with the nanoemulsion suspension to achieve a final concentration of 8% (vol/vol). All experiments were done in triplicate.

For *M. smegmatis*, a secondary culture was made by diluting a primary culture to an $OD_{600}$ of 0.5. A 500 µL aliquot of the suspension was placed in an autoclaved 1.7 mL microtube as described above, and biofilms were allowed to form for 5 days. The phytochemical treatment was conducted as described above for *M. abscessus*.

## CFU enumeration

Both planktonic and biofilm cultures were vortexed for 45 seconds and then sonicated in an ultrasonic bath (Branson) for 45 seconds to eliminate any mycobacterial clumping. A 25 µL aliquot of each culture was serially diluted in 225 µL complete Middlebrook 7H9 medium (as described above) from $10^0$ to $10^{-8}$. A 5 µL aliquot of each dilution was then spotted onto a Middlebrook 7H10 agar plate. Colonies were counted after 2 days for *M. smegmatis* and 3 days for *M. abscessus*.

## Lipid extraction and purification, and high-performance thin layer chromatography

Mycobacterial lipids were extracted, purified, and analyzed by high-performance thin layer chromatography as previously described (59). PIMs were visualized by orcinol staining.

## Wayne hypoxia model

*M. abscessus* log phase culture (OD$_{600}$ = 0.4–0.9) was diluted in a 1:100 ratio in 2 mL Middlebrook 7H9 broth supplemented as above plus 1.5 µg/mL methylene blue. These cells were subjected to slow withdrawal of oxygen as described earlier (66) over 5 days in an airtight sealed glass vial. On day 5, when methylene blue was completely reduced, we administered 3.1 mM eugenol, 8% E-BNE, or 28.8 µg/mL linezolid to the vials using a 27-gauge syringe. At both 0- and 48-hour time points post-treatment, 100 µL of culture was withdrawn from the airtight sealed vial using a 27-gauge syringe. A 25 µL aliquot of each culture was serially diluted in 225 µL Middlebrook 7H9 supplemented as above from $10^0$ to $10^{-8}$. A 5 µL aliquot of each dilution was then spotted onto a dried Middlebrook 7H10 agar plate. Colonies were counted after 4 days of incubation at 37°C.

## Flow cytometry of TO-PRO-3-stained cells

Two milliliter aliquots of *M. abscessus* log phase culture (OD$_{600}$ = 0.4–0.9) were transferred into a microtube and treated with 100 mM benzyl alcohol, 3.1 mM carvacrol, 3.1 mM eugenol, or 256 mM DMSO (vehicle control) for 1 hour at 37°C with shaking. After the treatment, the cells were stained with 0.1 mM TO-PRO-3 (Thermo Fisher Scientific) for 15 min at 37°C. Cells were centrifuged at 2,000 × *g* for 5 min at room temperature, and the pellet was resuspended in 1 mL of 2% formaldehyde in PBS. The fixed bacterial cells were centrifuged at the same condition, and the cell pellet was resuspended in PBS. The cells were analyzed by the LSRFortessa Cell Analyzer (BD Biosciences), and TO-PRO-3 fluorescence was detected using excitation wavelength at 640 nm and an emission bandpass filter (670/30 nm), following our previously established protocol (75).

## Mouse skin infection of *M. abscessus*

### Ethics

C57BL/6 mice (Jackson Laboratory) were housed in sterile cages with a 12-hour light/12-hour dark cycle. Mice were allowed to acclimatize for at least a week before any of the procedures were performed. We used three mice per group based on expected efficacy compared with previous studies (76), ensuring the ethical use of animals while obtaining statistically significant results.

### Generation and treatment of biofilm-infected murine skin wound

Mice were anesthetized using isoflurane, and meloxicam was subcutaneously administered for pain management. The skin on the dorsum of the mouse was shaved and disinfected with alternating povidone-iodine and alcohol swabs, thrice. Subsequently, a sterile 5-mm circular full-thickness skin wound was created using a skin puncture biopsy tool (Acuderm Inc., Fort Lauderdale, FL, USA). Using a micropipette, $10^8$ CFU

of luciferase-expressing *M. abscessus* in saline (10 µL) was inoculated onto the wound bed. To prevent secondary bacterial contamination and allow visualization of the wound bed, semi-occlusive transparent Tegaderm (3M, St. Paul, MN, USA) was affixed over the wound using Vetbond. Biofilm was allowed to form and mature for 4 days to simulate mature wound biofilm conditions. Infection was tracked through IVIS imaging of the luminescence signal from the bacteria. Then, mice were separated into three groups of three to receive one of the following: (i) PBS, (ii) 85 µM linezolid, and (iii) 8% E-BNE. The treatment was administered topically, once a day for 3 days. On the last day of treatment, the mice were sacrificed through $CO_2$ asphyxiation. A 3-mm circular full-thickness skin sample from the inner infection area was collected using a skin biopsy punch and then homogenized for quantitative bacterial colony counting.

## Statistical analysis

Statistical analysis was conducted using GraphPad Prism 10. As indicated in figure legends, one-way or two-way analysis of variance with a *post hoc* test was applied to data with a normal distribution. Statistical significance was set at *$P < 0.05$ and **$P < 0.005$.

## ACKNOWLEDGMENTS

We thank Eric Rubin (Harvard University) for sharing his luciferase-expressing *M. abscessus* strain.

This work was supported by the University of Massachusetts IALS Midigrant to Y.S.M. and V.M.R. V.M.R. acknowledges additional support from the NIH (R01 AI134770). M.P. was supported by a fellowship from the University of Massachusetts Amherst as part of the Chemistry-Biology Interface Training Program (National Research Service Award T32 GM008515 and GM139789). T.K. was supported by a postdoctoral fellowship from the Uehara Memorial Foundation. The microscopy data were gathered in the Light Microscopy Facility and Nikon Center of Excellence at the University of Massachusetts Institute for Applied Life Sciences.

## AUTHOR AFFILIATIONS

[1]Department of Microbiology, University of Massachusetts, Amherst, Massachusetts, USA
[2]Department of Chemistry, University of Massachusetts, Amherst, Massachusetts, USA

## PRESENT ADDRESS

Wyatt C. Tran, Department of Biochemistry and Molecular Pharmacology, New York University Grossman School of Medicine, New York, New York, USA
Takehiro Kado, Department of Biology, Missouri State University, Springfield, Missouri, USA

## AUTHOR ORCIDs

Casey Albano  http://orcid.org/0009-0003-5777-3493
Yasu S. Morita  http://orcid.org/0000-0002-4514-9242

## FUNDING

| Funder | Grant(s) | Author(s) |
|---|---|---|
| HHS \| NIH \| National Institute of Allergy and Infectious Diseases (NIAID) | R01 AI134770 | Vincent M. Rotello |

## ETHICS APPROVAL

All animal experiments were performed following the authorized protocol (IACUC Protocol ID 4022) and the policies issued by the Institutional Animal Care and Use Committee at the University of Massachusetts Amherst.

## ADDITIONAL FILES

The following material is available online.

### Supplemental Material

**Supplemental material (Spectrum02166-24-s0001.pdf).** Fig. S1 and S2.

### Open Peer Review

**PEER REVIEW HISTORY (review-history.pdf).** An accounting of the reviewer comments and feedback.

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
