## [Reviewer comments · Microbiology Spectrum]

Microbiology Spectrum

Effective killing of *Mycobacterium abscessus* biofilm by nanoemulsion delivery of plant phytochemicals

Casey Albano, Ahmed Nabawy, Wyatt Tran, Malavika Prithviraj, Takehiro Kado, Muhammad Hassan, Jessa Makabenta, Vince Rotello, and Yasu Morita

Corresponding Author(s): Yasu Morita, University of Massachusetts Amherst

Review Timeline:

Submission Date:	September 15, 2024
Editorial Decision:	October 10, 2024
Revision Received:	November 24, 2024
Accepted:	December 6, 2024

Editor: Olivier Neyrolles

Reviewer(s): Disclosure of reviewer identity is with reference to reviewer comments included in decision letter(s). The following individuals involved in review of your submission have agreed to reveal their identity: John Jairo Aguilera-Correa (Reviewer #1)

Transaction Report:

DOI: <https://doi.org/10.1128/spectrum.02166-24>

Re: Spectrum02166-24 (Effective killing of *Mycobacterium abscessus* biofilm by nanoemulsion delivery of plant phytochemicals)

Dear Dr. Morita,

Thank you for the privilege of reviewing your work. Below you will find instructions from the Spectrum editorial office, and the reviewer comments.

The two experts who reviewed your manuscript are enthusiastic about your study, and I encourage you to address their comments before we move forward.

Revision Guidelines

Sincerely,
Olivier Neyrolles
Editor
Microbiology Spectrum

Reviewer #1 (Comments for the Author):

The manuscript evaluates the efficacy of biodegradable polymeric nanoemulsions (BNE) loaded with plant-derivate oils against *M. abscessus* on both planktonic state and biofilm. Furthermore, such efficacy is proven in vivo in a murine in vivo model. The approach possesses novelty but the methodology and the way the results are expressed must be improved. Bellow, I propose

several comments:

Major comments

Regarding the biofilm grown on the air-liquid interface. Please, add some pictures of each type of biofilm (*M. smegmatis*, *M. abscessus*, untreated one, and treated ones) as a supplemental material.

Figure 7. An explanation of why C-BNEs are effective only at 6 days should be added. I would like to see how *M. abscessus* biofilms grow in the presence of unloaded BNE. It could be that that late C-BNE effect results from a slower metabolism of mycobacteria living inside that biofilm. Are the authors sure that mycobacteria cannot be used as a carbon source for the component of the BNE? Please, provide these results.

Regarding the in vivo model, based on what did the authors choose to use only three mice per group? Please, provide the sample size estimation and its statistical power.

Figure 8. The best in vitro results against *M. abscessus* was associated with C-BNE, why did the authors use E-BNE? This is the biggest scientific inconsistency of this study.

Figure 8b. It is a pity that the authors haven't analyzed the images taken with the IVIS® Spectrum Optical Imaging Platform. This platform allows us to quantify in vivo the infection progression without the need to take samples from the infected tissue. Do the authors have those data? If so, please, add them. If not, please, add this as a limitation of your study.

Figure 8c. I don't understand why the authors quantified the CFUs on the skin if the IVIS platform is more accurate. CFUs count in these models are less accurate and erratic. Anyway, these data must be treated as non-normally distributed data due to there are only three points per group, for that, they must be represented using median and interquartile range, and a Wilcoxon test must be applied. Furthermore, pictures are not consistent with the numerical data, the third mouse treated with E-BNE at day 7 seems to not have any luminescence in its loin.

Due to the Discussion section being short, I would recommend merging the Results section and the Discussion section.

Minor comments

The Introduction section lacks a paragraph describing the aims of the study. Please, add it.

Figure 2c must be replaced with another micrography with better quality. Some nanoemulsions can be irregular, but, in this case, they don't even look like spheroids.

Figure 6. Why are there no error bars?

In the Materials and Methods section, please, provide the morphotype of *M. abscessus* ATCC19977 and consider the use of only one *M. abscessus* morphotype in the Discussion sections as a limitation of your study.

A Statistical analyses section should be included at the end of the Materials and Methods section. Please, reflect the normality of your data and apply accordingly the corresponding statistical test.

Reviewer #2 (Comments for the Author):

The authors present an interesting study to determine if nanoemulsion delivery of phytochemicals is a viable delivery method for killing *Mycobacterium abscessus*. They have shown in vitro and in vivo activity and their conclusions are well supported. On the whole the paper is clear and well laid out, but there are a couple of areas that need addressing.

1. The data on killing of *Msmegmatis* and *Mabscessus* do not include a control of the nanoemulsion without the phytochemical.
2. The studies were conducted in triplicate, but the authors do not note if the error bars are standard deviation or standard error (they should be the former).

The manuscript evaluates the efficacy of biodegradable polymeric nanoemulsions (BNE) loaded with plant-derivate oils against *M. abscessus* on both planktonic state and biofilm. Furthermore, such efficacy is proven in vivo in a murine in vivo model. The approach possesses novelty but the methodology and the way the results are expressed must be improved. Bellow, I propose several comments:

Major comments

Regarding the biofilm grown on the air-liquid interface. Please, add some pictures of each type of biofilm (*M. smegmatis*, *M. abscessus*, untreated one, and treated ones) as a supplemental material.

Figure 7. An explanation of why C-BNEs are effective only at 6 days should be added. I would like to see how *M. abscessus* biofilms grow in the presence of unloaded BNE. It could be that that late C-BNE effect results from a slower metabolism of mycobacteria living inside that biofilm. Are the authors sure that mycobacteria cannot be used as a carbon source for the component of the BNE? Please, provide these results.

Regarding the in vivo model, based on what did the authors choose to use only three mice per group? Please, provide the sample size estimation and its statistical power.

Figure 8. The best in vitro results against *M. abscessus* was associated with C-BNE, why did the authors use E-BNE? This is the biggest scientific inconsistency of this study.

Figure 8b. It is a pity that the authors haven't analyzed the images taken with the IVIS® Spectrum Optical Imaging Platform. This platform allows us to quantify in vivo the infection progression without the need to take samples from the infected tissue. Do the authors have those data? If so, please, add them. If not, please, add this as a limitation of your study.

Figure 8c. I don't understand why the authors quantified the CFUs on the skin if the IVIS platform is more accurate. CFUs count in these models are less accurate and erratic. Anyway, these data must be treated as non-normally distributed data due to there are only three points per group, for that, they must be represented using median and interquartile range, and a Wilcoxon test must be applied. Furthermore, pictures are not consistent with the numerical data, the third mouse treated with E-BNE at day 7 seems to not have any luminescence in its loin.

Due to the Discussion section being short, I would recommend merging the Results section and the Discussion section.

Minor comments

The Introduction section lacks a paragraph describing the aims of the study. Please, add it.

Figure 2c must be replaced with another micrography with better quality. Some nanoemulsions can be irregular, but, in this case, they don't even look like spheroids.

Figure 6. Why are there no error bars?

In the Materials and Methods section, please, provide the morphotype of *M. abscessus* ATCC19977 and consider the use of only one *M. abscessus* morphotype in the Discussion sections as a limitation of your study.

A Statistical analyses section should be included at the end of the Materials and Methods section. Please, reflect the normality of your data and apply accordingly the corresponding statistical test.

Reviewer #1 (Comments for the Author):

The manuscript evaluates the efficacy of biodegradable polymeric nanoemulsions (BNE) loaded with plant-derivate oils against *M. abscessus* on both planktonic state and biofilm. Furthermore, such efficacy is proven in vivo in a murine in vivo model. The approach possesses novelty but the methodology and the way the results are expressed must be improved. Bellow, I propose several comments:

We thank the reviewer for acknowledging the novelty of our research and providing constructive suggestions. We addressed the comments as detailed below.

Major comments

Regarding the biofilm grown on the air-liquid interface. Please, add some pictures of each type of biofilm (*M. smegmatis*, *M. abscessus*, untreated one, and treated ones) as a supplemental material.

We have added the images of the biofilm as requested (see **Supplemental Figure S1**). The pellicle becomes disrupted when we drain the medium and replace with a fresh medium containing BNE. However, the overall appearance of the pellicle does not change significantly with or without BNE treatments. E-BNE treatments are representative, and are the only condition shown. We describe these results in the revised manuscript (see **Lines 168-171 and 240-242**).

Figure 7. An explanation of why C-BNEs are effective only at 6 days should be added. I would like to see how *M. abscessus* biofilms grow in the presence of unloaded BNE. It could be that that late C-BNE effect results from a slower metabolism of mycobacteria living inside that biofilm. Are the authors sure that mycobacteria cannot be used as a carbon source for the component of the BNE? Please, provide these results.

As the reviewer suggested, we speculate that it takes time to kill mycobacterial biofilm because it is slow for the carvacrol to reach the inside of the biofilm even when C-BNE is used. We have added text addressing this issue (**Lines 247 - 248**).

Regarding the possibility of BNE as a carbon source, the polymer is in low concentrations in the nanoemulsion (330 µg/ml of 100 % BNE), and the BNE is likewise in low concentration relative to other nutrients in the media. As such, the antimicrobial is not serving as a significant nutrient during treatment. We discuss these considerations in the Results and Discussion (**Lines 248 - 251**).

Regarding the in vivo model, based on what did the authors choose to use only three mice per group? Please, provide the sample size estimation and its statistical power.

We used three mice per group (n=3) because this number provided sufficient statistical power to detect a significant difference between the control and untreated groups. This sample size was based on previous studies (Pandya AN, Antimicrob. Agents Chemother. 2019, 63, e02245-18) and ensures the ethical use of animals while obtaining statistically significant results. We added the justification in the Methods section (**Lines 412 – 414**).

Figure 8. The best in vitro results against *M. abscessus* was associated with C-BNE, why did the authors use E-BNE? This is the biggest scientific inconsistency of this study.

We thank the reviewer for this comment. As the reviewer pointed out, E-BNE was less effective than C-BNE in our in vitro studies. However, we have previously shown that E-BNE is less toxic to 3T3 fibroblast cells than C-BNE. Therefore, we used E-BNE to test its efficacy against dormant *M. abscessus* as well as *M. abscessus* in a wound infection model. We have discussed this point in the Discussion in the initially submitted manuscript.

In the revised manuscript, we combined the Results and Discussion sections in response to the suggestion by Reviewer 1. Therefore, we moved this justification to the section where we described Figure 8 to reiterate our rationale (**Lines 256 – 260**).

Figure 8b. It is a pity that the authors haven't analyzed the images taken with the IVIS® Spectrum Optical Imaging Platform. This platform allows us to quantify in vivo the infection progression without the need to take samples from the infected tissue. Do the authors have those data? If so, please, add them. If not, please, add this as a limitation of your study.

Thank you for the suggestion. We have now included the image analysis data, which shows a similar pattern to the colony counts obtained from isolated tissue samples from the infected area, as seen in **Supplemental Figure S2**.

Figure 8c. I don't understand why the authors quantified the CFUs on the skin if the IVIS platform is more accurate. CFUs count in these models are less accurate and erratic. Anyway, these data must be treated as non-normally distributed data due to there are only three points per group, for that, they must be represented using median and interquartile range, and a Wilcoxon test must be applied. Furthermore, pictures are not consistent with the numerical data, the third mouse treated with E-BNE at day 7 seems to not have any luminescence in its loin.

The IVIS imaging system is a useful tool for tracking live infections, but it is less sensitive and accurate for quantifying bacterial infection at the wound site. Figure 8B shows that no bacterial signal was detected on day 7 post-treatment in IVIS images. However, Figure 8C demonstrates the presence of a residual bacterial population ($\sim 10^4$ - 10^5 CFU/g) in tissue samples collected after euthanizing the mice on day 7. Overall, IVIS provides a useful tool, however colony counting is more sensitive and accurate than IVIS for quantification of bacterial load.

We discussed this point in the revised manuscript (**Lines 270 – 274**), and indicated a reference that supports our point (van Oosten, M.; Hahn, M.; Crane, L.M.A.; Pleijhuis, R.G.; Francis, K.P.; van Dijk, J.M.; van Dam, G.M. Targeted Imaging of Bacterial Infections: Advances, Hurdles and Hopes. *FEMS Microbiol. Rev.* 2015, 39, 892-916).

Due to the Discussion section being short, I would recommend merging the Results section and the Discussion section.

We have combined the Results and Discussion sections as suggested.

Minor comments

The Introduction section lacks a paragraph describing the aims of the study. Please, add it.

We have added a sentence as suggested (**Lines 124-125**).

Figure 2c must be replaced with another micrograph with better quality. Some nanoemulsions can be irregular, but, in this case, they don't even look like spheroids.

The oil component of nanosponges evaporates when the sample is prepared for electron microscopy. As such, the resulting micrographs of nanosponges are irregular, analogous to deflated balloons. We added this explanation in the figure legend (**Lines 696 – 697**).

Figure 6. Why are there no error bars?

As requested, we have changed the graph to show the averages and standard deviations.

In the Materials and Methods section, please, provide the morphotype of *M. abscessus* ATCC19977 and consider the use of only one *M. abscessus* morphotype in the Discussion sections as a limitation of your study.

ATCC19977 is a smooth morphotype strain. We added this information in the revised manuscript (**Lines 341 – 342**).

We agree that the use of one morphotype of *M. abscessus* is a limitation of our study, and we plan to explore the effect of BNE-mediated drug delivery against rough morphotype strains in the future. We added this discussion in the revised manuscript (**Lines 290 – 292**).

A Statistical analyses section should be included at the end of the Materials and Methods section. Please, reflect the normality of your data and apply accordingly the corresponding statistical test.

As requested, a statistical analysis section was added at the end of the Materials and Methods section (**Lines 433 – 436**).

Reviewer #2 (Comments for the Author):

The authors present an interesting study to determine if nanoemulsion delivery of phytochemicals is a viable delivery method for killing *Mycobacterium abscessus*. They have shown in vitro and in vivo activity and their conclusions are well supported. On the whole the paper is clear and well laid out, but there are a couple of areas that need addressing.

We thank the reviewer for positive comments. Our responses to the reviewer's comments are shown below.

1. The data on killing of *Msmegmatis* and *Mabscessus* do not include a control of the nanoemulsion without the phytochemical.

We appreciate the reviewer's comment. However, the oil component is central to the nanoemulsion structure and activity. The polymer alone cannot be emulsified and has very different properties. Hence, it is not a very useful control. We added this note in the legend for Figure 3 (**Lines 708 – 709**).

2. The studies were conducted in triplicate, but the authors do not note if the error bars are standard deviation or standard error (they should be the former).

As the reviewer pointed out, our error bars are standard deviations. We have added this information in the figure legends (**Lines 725, 730 – 731, 740, 746**).

Re: Spectrum02166-24R1 (Effective killing of *Mycobacterium abscessus* biofilm by nanoemulsion delivery of plant phytochemicals)

Dear Dr. Yasu S. Morita:

Your manuscript has been accepted, and I am forwarding it to the ASM production staff for publication. Your paper will first be checked to make sure all elements meet the technical requirements. ASM staff will contact you if anything needs to be revised before copyediting and production can begin. Otherwise, you will be notified when your proofs are ready to be viewed.

Sincerely,
Olivier Neyrolles
Editor
Microbiology Spectrum